# Retinoic Acid Improves the Recovery of Replication-Competent Virus from Latent SIV Infected Cells

**DOI:** 10.3390/cells9092076

**Published:** 2020-09-11

**Authors:** Omalla A. Olwenyi, Arpan Acharya, Nanda Kishore Routhu, Keely Pierzchalski, Jace W. Jones, Maureen A. Kane, Neil Sidell, Mahesh Mohan, Siddappa N. Byrareddy

**Affiliations:** 1Department of Pharmacology and Experimental Neuroscience, University of Nebraska Medical Center, Omaha, NE 68198-5800, USA; omalla.olwenyi@unmc.edu (O.A.O.); arpan.acharya@unmc.edu (A.A.); nanda.kishore.routhu@emory.edu (N.K.R.); 2Department of Pathology and Microbiology, University of Nebraska Medical Center, Omaha, NE 68198-5800, USA; 3Department of Pharmaceutical Sciences, University of Maryland School of Pharmacy, Baltimore, MD 21201, USA; keelski1@gmail.com (K.P.); jjones@rx.umaryland.edu (J.W.J.); mkane@rx.umaryland.edu (M.A.K.); 4Department of Obstetrics and Gynecology, Emory University School of Medicine, Atlanta, GA 30322, USA; nsidell@emory.edu; 5Texas Biomedical Research Institute, Southwest National Primate Research Institute, San Antonio, TX 78227, USA; mmohan@txbiomed.org; 6Department of Genetics, Cell Biology and Anatomy, University of Nebraska Medical Center, Omaha, NE 68198-5800, USA; 7Department of Biochemistry and Molecular Biology, University of Nebraska Medical Center, Omaha, NE 68198-5800, USA

**Keywords:** immune activation, retinoic acid, CD4 T cells, PBMCs, SIV, viral load and provirus, CD3/CD28 treatment

## Abstract

The accurate estimation and eradication of Human Immunodeficiency Virus (HIV) viral reservoirs is limited by the incomplete reactivation of cells harboring the latent replication-competent virus. We investigated whether the in vitro and in vivo addition of retinoic acid (RA) enhances virus replication and improves the detection of latent virus. Peripheral blood mononuclear cells (PBMCs) from naive and anti-retroviral therapy (ART)-treated SIV-infected rhesus macaques (RMs) were cultured in vitro with anti-CD3/CD28 + IL-2 in the presence/absence of RA. Viral RNA and p27 levels were quantified using RT-qPCR and ELISA, respectively. Viral reservoirs were estimated using the Tat/Rev-Induced Limited Dilution Assay (TILDA) and Quantitative Viral Outgrowth Assay (QVOA). In vitro and in vivo measures revealed that there was also an increase in viral replication in RA-treated versus without RA conditions. In parallel, the addition of RA to either CD3/CD28 or phorbol myristate acetate (PMA)/ionomycin during QVOA and TILDA, respectively, was shown to augment reactivation of the replication-competent viral reservoir in anti-retroviral therapy (ART)-suppressed RMs as shown by a greater than 2.3-fold increase for QVOA and 1 to 2-fold increments for multi-spliced RNA per million CD4^+^ T cells. The use of RA can be a useful approach to enhance the efficiency of current protocols used for in vitro and potentially in vivo estimates of CD4^+^ T cell latent reservoirs. In addition, flow cytometry analysis revealed that RA improved estimates of various viral reservoir assays by eliciting broad CD4 T-cell activation as demonstrated by elevated CD25 and CD38 but reduced CD69 and PD-1 expressing cells.

## 1. Introduction

Of the 37.9 million individuals living with Human Immunodeficiency Virus (HIV) worldwide, probable viral eradication has only been achieved in two people: “the Berlin and the London patients” [1,2]. Subsequent attempts have failed to replicate a sterilizing cure in several follow up studies [3,4]. Hence, most researchers have focused on the reduction and/or elimination of “viral reservoirs” to achieve a functional cure capable of controlling viremia in the absence of continued anti-retroviral therapy (ART). However, the major roadblock facing functional cure strategies is that there is no robust method that can accurately quantify viral reservoirs in various host anatomical compartments [5]. Furthermore, there is a poor understanding of the cellular lineages that comprise the reservoir [6,7].

Presently, the gold standard for approximating the size of the replication-competent viral reservoir is the Quantitative Viral Outgrowth Assay (QVOA). However, it yields a minimal estimate since a fraction of latent cells requires multiple rounds of activation to reproduce functional virions [8]. In addition, proviral sequencing has not only reported over 5-fold levels of the size of the viral reservoir in comparison to QVOA but has also queried its estimates in different CD4+ T cell subsets. While QVOA measures have shown central memory CD4+ T cells to contain the largest fraction of the reservoir [9], proviral sequencing also revealed that intact pro-viruses were predominantly found in effector memory T cells (TEM) > transitional memory (TTM) > naïve > central memory T cells (TCM) [10]. Induced virion assays (IVA) that yield close to similar estimates as QVOA have also been developed [11]. Alternatively, proviral DNA based assays like the Alu Gag PCR, Total HIV-1 DNA quantification PCR and cell-associated RNA produce overestimates, as most of the integrated viral genome is defective [12,13,14,15]. Similarly, the Tat/Rev-Induced Limited Dilution Assay (TILDA) also provides overestimates of the size of the functional reservoir as all cells producing multi-spliced RNAs (msRNA) may not have the potential to produce infectious viral particles [16]. At a single cell level, transcription/translation measures of the viral reservoir can also be performed. This involves using intra cellular flow cytometry to detect the HIV core antigen (p24) and in-situ hybridization for co-expression of HIV RNA. However, this approach also provides overestimates of the viral reservoir [17].

Several attempts have been undertaken to improve these assays. Recently, Burner et al., and Bender et al., sought to enhance the sensitivity of PCR assays used for the quantification of the viral reservoir. To do this, they carried out whole-genome sequencing of ART-treated HIV-1, HIV-2 and Simian Immunodeficiency Virus (SIV) in order to evaluate the landscape of the viral reservoir. Then, they carefully developed primer probe sets targeting two frequently mutated regions that, when used in combination in a digital droplet PCR assay, could discriminate and quantify both intact and defective HIV-1 provirus/SIV/SHIV and HIV-2 infections [18,19]. Since this assay does not interrogate the whole genome, other deletions or hypermutations could be omitted and later affect the accuracy of its measurements. The sensitivity of QVOA has also recently been improved by enhancing virus outgrowth through the adoptive transfer of HIV-1 latent cells into humanized mice [20].

Recently, Zhang et al., 2020, showed that QVOA estimates could be improved by refining cell culture conditions after T cell receptor (TCR) stimulation of sorted memory CD4^+^ T cells and later supplementation with retinoic acid (RA) for improved viral outgrowth [21]. However, it remains to be determined whether the supplementation with RA could be modified to improve QVOA estimates in latently-infected cells obtained in vivo from non-human primate (NHP) models such as rhesus macaques (RMs). In addition, the full spectrum of the potential of RA in re-activating the viral reservoir in other assays besides QVOA remains poorly understood. Since the viral reservoir is highly heterogenous [22,23,24,25], the utility of RA supplementation to activate total CD4^+^ T cells as opposed to sorted memory CD4^+^ T cells remains unknown. Finally, the step by step evaluation of how the addition of RA to CD3/CD28-stimulated cells orientates the CD4^+^ T cell phenotype towards improved re-activation of the viral reservoir is yet to be fully comprehended.

Based on these observations, we reasoned that the ex-vivo addition of RA to CD4^+^ T cells from SIV-infected ART-suppressed RMs could be used to improve the reactivation of the latent reservoir following T cell receptor (TCR) activation in multiple assays such as QVOA and TILDA. We also tested whether RA was capable of activating virus replication in SIV infected RMs. This was followed by sequential profiling of accompanying immunophenotypic changes that occur after the addition of RA to TCR-stimulated CD4^+^ T cells. The generated findings highlight potential processes by which improved reservoir reactivation could occur following the addition of RA.

In this study, we demonstrate that the addition of RA to CD3/CD28 or phorbol myristate acetate (PMA)/Ionomycin activated CD4^+^ T cells obtained from RMs improved activation of the latent reservoir in multiple assays such as QVOA and TILDA. The α4β7^+^ fraction had distinct immune activation profiles as demonstrated by reduced expression of the early immune activation marker CD69 [26] and the terminal immune activation/exhaustion marker PD-1 [27]. Instead, mid-stage immune activation markers such as CD25 and CD38 [28,29] were elevated in the α4β7^+^ compared to the α4β7^−^ fraction of TCR stimulated cells. As expected, the addition of RA led to enhanced production of viral particles in ex-vivo cultures of peripheral blood mononuclear cells (PBMCs) obtained from ART-treated SIV-infected macaques. These observations were translated in an in vivo model where the administration of RA to chronic SIV infected macaques was associated with non-correlating trends of increasing viral load. In tandem, there was a subsequent expansion of naive CD4^+^ T cells.

Herein, we submit that a refinement of current QVOA/TILDA reservoir protocols through the addition of RA can potentially be utilized to improve measures of latently SIV-infected cells obtained from RMs. RA supplementation augments QVOA/TILDA readouts by favoring immunophenotypic changes characterized by increased α4β7^+^ expression. This is accompanied by sustained mid-stage immune activation with minimal plateauing into terminal activation/immune exhaustion. Lastly, we also document the ability of RA to enhance virus replication in vivo and show its utility in future experiments that would seek to improve the readouts for diverse latent reservoir assays using NHP models such as RMs.

## 2. Materials and Methods

### 2.1. Source of Blood Samples and Tissues

Heparinized blood samples were collected from RMs (*Macaca mulatta*), housed at three separate locations. These comprised of the Yerkes National Primate Research Center (YNPRC) of Emory University, University of Nebraska Medical Center (UNMC) and the Tulane National Primate Research Center (TNPRC). Tissue (Blood and lymph nodes) sampling was carried out on SIV naive or SIVmac239/SIVmac251 infected RMs. Some of the SIV infected RMs were treated with combined anti-retroviral therapy (cART) resulting in a marked reduction in plasma viral loads (<50 copies/mL) as described in Appendix A. For the in vivo studies, SIVmac239 chronically infected RMs with low (ROg13 and RYd13) and high (RPs13 and RZc14) plasma viral loads were utilized. Protocols for the studies conducted at Emory University were reviewed and approved by the Emory University IACUC and assigned the IACUC protocol number “YER-2001725-042715GA”. Studies conducted at UNMC were reviewed and approved by the UNMC IACUC protocol 15-073 “Role of HIV Env glycosylation in mucosal transmission.” Alternatively, studies performed at TNPRC were approved by the TNPRC Institutional Animal Care and Use Committee (Protocol number 3781). The housing, care, diet, and maintenance conformed with the guidelines of the Committee on the Care and Use of Laboratory Animals of the Institute of Laboratory Animal Resources.

### 2.2. Simian Immunodeficiency Virus (SIV) Replication, Virus Expansion and Latent Reservoir Measurement Assays

#### 2.2.1. SIV Replication Assays

In an effort to determine the effects of RA on in vitro viral replication, PBMCs were obtained from ten (*n* = 10) randomly selected uninfected RMs and cultured in (a) RPMI 1640 supplemented with 20% heat-inactivated Fetal Bovine Serum, l-Glutamine (2 mM) and Penicillin (100 units/mL)/Streptomycin (100 μg/mL)) alone (media control), (b) media + anti CD3/CD28 beads (c) media + anti CD3/CD28 + IL-2 and d) media + anti CD3/CD28 + IL-2 + RA. After 72 h, cultured PBMCs were infected with SIVmac239 (grown in day 3 Con-A activated normal rhesus PBMC cultures). On day 10 post-infection, the culture supernatants were assayed for levels of p27 using an ELISA kit (Advanced Bioscience Laboratories (ABL) inc, Rockville, MD, USA).

#### 2.2.2. Virus Expansion Assays

For virus expansion assays, infected CD4^+^ T cells were isolated from PBMCs (pooled from ten macaques, as described above) using the CD4^+^ T cell isolation kit (Miltenyi Biotec, Auburn, CA, USA) and cells treated under similar conditions as described in the earlier experiment. After this, the CD4^+^ T cells were counted and 2 × 10^6^ cells plated across each well of a 12 well Nunc cell culture plate (Thermo Scientific Auburn, AL, USA). Thereafter, co-culture experiments were carried out. This involved the addition of CEMX174 cells added at respective serial dilutions (5%, 10%, 25%, 50%, 75%, 100% CEMX174 co-cultures). Notably, the lowest dilution (100% CEMX174) comprised of 2 × 10^6^ CEMX174 cells while the highest dilution (5% CEMX174) contained 2.5 × 10^5^ CEMX174 cells. Following this, the levels of p27 were measured at either day 7 or day 10.

#### 2.2.3. Latent Reservoir Quantification Assays

Tat/Rev-Induced Limited Dilution Assay (TILDA)

Next, we performed modified TILDA using auxiliary lymph nodes and PBMCs obtained from four and five ART-treated macaques respectively (Appendix A). This involved the addition of 1 µM of RA following the activation of enriched CD4^+^ T cells using 100 ng/mL of phorbol myristate acetate (PMA) and 1 μg of ionomycin. After activation different CD4^+^ T cell dilutions (18,000, 9000, 3000 and 1000 cells/well) were made. Later on, using an ultra-sensitive nested PCR assay, the production of Tat/rev multiply spliced transcripts were quantified and the values expressed as Tat/rev msRNA per million CD4^+^ T cells as described in detail elsewhere [30,31].

Quantitive Viral Outgrowth Assay (QVOA)

Finally, for QVOA, CD4^+^ T cells were negatively selected from PBMCs using a non-human primate microbead CD4^+^ T cell isolation kit, (STEMCELL Technologies Inc, Seattle, WA, USA). The resulting CD4^+^ T-cell population possessed greater than 95% purity. Thereafter, the purified CD4^+^ T cells were counted and resuspended in media containing RPMI1640, 10% FBS, 2 mM glutamine, Penstrep (100 U/mL penicillin and 100 μg/mL streptomycin) and 10 U/mL IL-2. Following this, the suspension culture media was then supplemented with 300 nM efavirenz (EFV; NIH AIDS Reagent Program) so as to limit multiple rounds of SIV infection [32,33]. Anti-human CD3/CD28 mAb-coated microbeads (Dynabeads, Life Technologies, Waltham, MA, USA) were then added at a 1:1 bead to T cell ratio and cultured for 3 days. Later on, two sets of 10-fold limiting dilutions ranging from 1 to 10^2^ × 10^5^ cells/mL respectively were performed. In one set of cells, 1 µM of RA was added in the dark after 24 h of activation. In the other set, no RA was added. After 72 h of incubation, 10^5^ CEMx174 cells were added in each well to expand the virus released from activated cells. This time point was noted as day 0 of incubation and was later prospectively followed up to 21 days. The cells were split at a 1:4 ratio on days 3, 7, 10, 14 and 17. Cell culture supernatants were collected on days 3, 7, 14 and 21 and analyzed for the levels of SIV RNA using quantitative real-time polymerase chain reaction (qRT-PCR). The frequency of cells harboring replication-competent viruses was estimated using the IUPMStats v1.0 infection frequency calculator. Thereafter, the final values were reported as infectious units per million (IUPM) [34].

### 2.3. Retinoic Acid (RA) Treatment of RMs and Measurement in Plasma

#### 2.3.1. Retinoic Acid (RA) Treatment of RMs

Chronically SIVmac239-infected RMs were treated orally with RA (10 mg/kg daily for 10 days). Following RA treatment (Day 0), RMs with either high (ROg13 and RYd13) or low (RPs13 and RZc14) plasma viral loads were prospectively monitored for changes in: levels of plasma viremia, total, naive, CM and EM CD4^+^ T cell subsets for the duration of 28 days. This in vivo model of RA treatment was chosen based on a previous study in which the authors sought to understand the pharmacokinetics of the gastric administration of this metabolite when given to Cynomolgus monkeys for 10 days [35]. Their findings showed that the daily administration of 10 mg/kg RA achieved peak plasma concentration of RA 2–4 h after dosing. This was later followed by a rapid decline of the metabolite. Furthermore, a ten-day duration of dosing was preferred because RA induces its own oxidative catabolism hence reducing its bioavailability after a few weeks of daily treatment [36]. As such, we limited our treatment protocol to a duration of 10 days as reported by Kraft et al., 1991 [35].

#### 2.3.2. Retinoic Acid (RA) Measurement in Plasma

Following 3 to 12 h after dosing, RA levels were measured in collected venous blood obtained by venipuncture after performing anesthesia of the RA-treated animals. Plasma obtained from the collected venous blood was stored at −80 °C until the time of extraction. Geometric isomers of RA were quantified from plasma according to a previously validated methodology [37]. The extraction of retinoids was performed using a two-step liquid-liquid extraction using 4,4-dimethyl-RA as an internal standard for RA [37,38]. After this, the levels of RA were determined by liquid chromatography-multi stage tandem mass spectrophotometry (LC-MRM^3^) that utilized two distinct fragmentation events for enhanced selectivity [37]. Briefly, a Shimadzu Prominence UFLC XR liquid chromatography system (Shimadzu, Colombia, MD, USA) coupled to an AB Sciex 5500 QTRAP hybrid triple quadrupole mass spectrometer (AB Sciex, Framingham, MA, USA) that utilizes atmospheric pressure chemical ionization (APCI) whilst operating in positive ion mode was used [37]. The amount of RA was normalized per ml and data were expressed as mean ± standard deviation for three completely independent determinations per NHP per timepoint. Furthermore, prospective changes in viral loads and tandem changes in memory/differentiation status of different CD4^+^ T cell subsets following the administration of RA were also investigated.

### 2.4. Isolation of Cells from Blood/Tissues

Mononuclear cells were isolated from blood and lymph node biopsy specimens from RMs as previously reported [39]. Briefly, PBMCs were isolated from collected heparinized blood using Ficoll–Hypaque density gradient centrifugation. For lymph nodes, 10% complete media (RPMI 1640 medium supplemented with 10% heat-inactivated bovine calf serum, 2 mM L-glutamine, 25 mM HEPES, penicillin and 1% streptomycin (pen/strep)) was added to collected lymph node tissues. The lymph node biopsies were later chopped into smaller pieces and cell suspensions obtained through filtration with aid of 100 and 40 µm nylon mesh sterile cell strainers (Thermo Fischer, Waltham, MA, USA) respectively. Cell viability was then estimated using the Trypan Blue exclusion as previously described [40] and counts performed using the Countess II FL Automated Cell Counter (Invitrogen, Thermoscientific Waltham, MA, USA). The viabilities of cells generated from the processing of the lymph node and whole blood samples ranged from 70 to 95%.

### 2.5. Cell Culture

The CEMx174 hybrid cell line comprising of the human B 721.174 and human T CEM cell lines was used for virus expansion due to its increased susceptibility to SIV/HIV infection [41]. This cell line was originally provided by J. Hoxie (University of Pennsylvania, Philadelphia). Prior to use, these cells were cultured and maintained in 10% culture media (RPMI 1640 medium supplemented with 10% heat-inactivated, bovine calf serum, 2 mM l-glutamine, 25 mM HEPES, penicillin and 1% streptomycin (pen/strep)) at 37 °C and 5% CO_2_. For subsequent cell culture, media utilized consisted of RPMI 1640 supplemented with 10% heat-inactivated Fetal Bovine Serum, l-Glutamine (2 mM) and Penicillin (100 units/mL)/Streptomycin (100 μg/mL). Our study group and others have found that the 1 µM RA (Sigma Aldrich, St. Louis, USA) concentration is optimal for immune modulation of diverse leukocyte subsets whilst maintaining cell viability (>80%) [42,43]. Hence, PBMCs were cultured in a) media alone (control) b) media containing 10 U/mL of rhIL-2 (IL-2) (Roche, Basel, Switzerland) + 1 µM RA c) media containing anti-CD3/CD28 conjugated immunobeads (Life Sciences) + IL-2 or d) media containing anti-CD3/CD28 mAb-coated microbeads (Life Technologies, Carlsbad, USA) + IL-2 + 1 µM RA conditions. In detail, the cultures were incubated for the first 24 h at 37 °C and 5% CO_2_ with/without the addition of anti-CD3/CD28 beads at a bead to cell ratio of 1:1. Following the 24 h culture period, IL-2 and RA were added to the appropriate culture conditions.

### 2.6. Flow Cytometry

#### 2.6.1. Quantification of Diverse Immune Activation Markers

PBMCs collected from naive animals were utilized for the evaluation of the frequencies and density of α4β7^hi^ expression on CD4^+^ T cells using standard flow cytometry as described previously [42]. Similarly, the changes in immune activation (IA) and memory/differentiation cell markers on the gated population of α4β7^+^ versus α4β7^−^ CD4^+^ T cells were evaluated in CD3/CD28 + IL-2 and CD3/CD28 + IL-2 + RA conditions. IA markers were chosen to cover the different stages of this process. For early activation, we sought CD25 and CD69 whilst mid activation states were represented by CD38 and HLA-DR. The late stages of activation were tested using PD-1 and the exhaustion marker CTLA4. For the IA panel, isolated PBMCS were stained with a 1:1000 Zombie Aqua^TM^ Fixable viability solution (BioLegend, San Diego, CA, USA) and incubated for 30 min in the dark. The cells were washed with protein-free PBS and stained with anti- CD38 PE (non-human primate (NHP) reagent Resource), anti CD4 BV605 Clone L200 (BD optibuild, San Jose, CA, USA), anti-CD3 AF700 Clone SP34-2 (BD Biosciences), anti-Ki67 FITC Clone B56 (BD biosciences), anti-PD-1 PECy7 Clone EH12.2H7 9 (BioLegend, San Diego, CA, USA), anti-Human HLA-DR Class II PE Texas Red Clone MHLDR (Life Technologies/Thermoscientific Waltham, MA, USA), anti-CD25 Pac blue Clone BC96 (BioLegend) and anti CD69 APCH7 Clone FN50 (BD Biosciences). Thereafter, the cells were washed with PBS and fixed with 1% paraformaldehyde (PFA). Following this, a strict ten-minute permeabilization was performed using 1 × BD perm wash buffer (Catalogue number: 51-2091 KZ). This was followed by intracellular staining using anti-human CTLA4 Clone BN13.1 (BD biosciences).

#### 2.6.2. Delineation of Memory and Differentiation Status

For the memory and differentiation panel, similar steps were carried out during surface staining. This involved the addition of anti-CD10 APC Cy7 Clone H110a (BioLegend), anti-CD3 AF700 Clone SP34-2 (BD Biosciences), anti-CD4 BV605 Clone L200 (BD optibuild, BD Biosciences), anti-CD45RA PE-CF594 Clone 5H9 (BD Horizon, BD Biosciences), anti-human CCR7 PE Cy7 Clone 3D12 (BD Biosciences) monoclonal antibodies. Lastly, to determine the distribution of IA markers within the studied CD4^+^ T cell subsets, anti- CD38 PE (NHP reagent Resource) and anti-PD-1 PECy5.5 Clone EH12.2H7 9 (BioLegend) were added to the PE and PE Cy5.5 channels of the memory and differential panel and similar procedures for flow cytometry surface staining followed. In a separate panel, anti-CD28 APC Clone CD28.2 (BD Biosciences) and anti-CD95 PerCP-Cy^TM^ 5.5 Clone DX2 (BD Pharmingen^TM^, BD Biosciences) monoclonal antibodies were included. This panel was used to characterize memory and differentiation status in PBMCs obtained from SIV infected RMs treated with RA of blood. Following this, cells were fixed with 1% PFA.

#### 2.6.3. Data Acquisition and Analysis of Generated Flow Cytometry Data

All events were later acquired using the Attune Nxt flow cytometer or BD LSR FortessaX450. Generated Flow Cytometry Standard (FCS) files were later analyzed using Flow Jo Version 10.5 (Ashland, OR, USA). Fluorescence minus one or two (FMO/FM2) controls were used to set cut off gates during the quantitation of most markers in all flow cytometry experiments.

### 2.7. Statistical Analysis

Using Graph Pad Prism Version 7.0 C, differences within paired groups were obtained using Wilcoxon matched pairs signed rank tests. All values with *p* < 0.05 were considered statistically significant.

## 3. Results

### 3.1. Retinoic Acid (RA) Enhances SIV Replication and Improves Latent SIV Quantitation

Investigations were performed to evaluate the effect of RA on the SIV replication in vitro. To achieve this, PBMCs from 10 randomly selected uninfected RMs were stimulated with anti-CD3/CD28 beads alone, anti-CD3/CD28 beads + IL-2 and anti-CD3/CD28 beads + IL-2 + RA and then infected with SIVmac239 as described in Materials and Methods. As seen in Figure 1A, in cells harvested from 9 out of 10 RMs, there was a greater than threefold increase in p27 levels in culture supernatants of the anti-CD3/CD28 + IL-2 + RA-treated group compared to the anti-CD3/CD28 + IL-2 groups at day 10 (*p* < 0.0001), (Figure 1A). Furthermore, co-culture of CD4^+^ T cells (obtained from pooled PBMCs) with CEMX174 cells across a range of dilutions revealed a greater than 5-fold increase in p27 levels following the addition of RA to the anti-CD3/CD28 + IL-2-treated CD4^+^ T cells as compared to the anti-CD3/CD28 + IL-2 only treated CD4^+^ T cells at day 7 (*p* = 0.0144) and day 10 (*p* = 0.0235) respectively (Figure 1B). We next examined if RA treatment could enhance the activation of latent reservoir cells from SIV-infected ART-suppressed macaques. First, we performed a quantitative viral outgrowth assay (QVOA). Here, total CD4^+^ T cells were isolated from PBMCS obtained from combined antiretroviral therapy (cART)-treated SIVmac251 infected macaques (*n* = 8) with viral loads <50 copies/mL (Appendix A). The cells were treated with anti-CD3/CD28 beads with and without RA as described in the methods section. As shown in Figure 1C, we observed a 2.5-fold increase in reactivation of the replication-competent CD4^+^ T-cell viral reservoir in anti-CD3/CD28 + RA-treated animals (15.09 (3.56–105.3) IUPM) in comparison to the anti-CD3/CD28 alone group (6.38 (1.04–36.43) IUPM), (*p* < 0.01). Similarly, in TILDA assays performed using PBMC derived CD4^+^ T cells (*n* = 5), there was an upregulation in copies of msRNA/million CD4^+^ T cells upon RA treatment following PMA + ionomycin activation versus either PMA + ionomycin alone or the media control groups (*p* < 0.05), (Figure 1D). A similar trend was observed in the lymph node-derived CD4^+^ T cells obtained from (*n* = 4) RMs (Figure 1D). The addition of RA to PMA and Ionomycin during T cell activation resulted in greater recovery of msRNA/million CD4^+^ T cells compared with either PMA + ionomycin and unstimulated conditions.

### 3.2. Oral Administration of All Trans RA to SIV Infected RMs Coincides with Increase in Viral Loads within the High Viral Load Group

We tested in a pilot study whether RA impacted viral replication in chronically SIV infected RMs. Here, RA was orally administered daily for a duration of 10 days in chronically infected SIVmac239 infected macaques (*n* = 4) with either low viral loads (RPs13 and RZc14) or high viral loads (ROg13 and RYd13). Together with bio-available RA level measures and immune phenotyping of CD4^+^ T cell subsets, accompanying plasma viral loads were also monitored up to day 28 as shown in Figure 2A. All animals showed increased plasma RA levels to varying degrees at day 7 accompanied with a return to baseline at day 28 (Figure 2B). Viral load measures also revealed that viral set points were stable for more than 2 months prior to RA treatment. Upon RA administration, three animals (ROg13, RYd13 and RPs13), displayed increases in plasma viral loads on days 7–14 as compared to baseline which later stabilized upon withdrawal of the RA 10 days post-therapy (Figure 2C).

Interestingly, there was a trend of an increased bioavailability of RA coinciding with enhanced virus replication (Figure 2B,C). Clearly, RMs (ROg13 and RYd13) with higher levels of plasma RA displayed increased SIV replication kinetics. On the other hand, RMs with low levels of plasma RA (RPs13 and RZc14) had slight and no changes in viral replication during the RA treatment phase, respectively (Figure 2C). Following the oral administration of RA, there was an expansion of α4β7^+^ CD4^+^ T cells expressing a predominantly naive cell phenotype (Figure 2D). Peak increases of α4β7^+^ CD4^+^ T cells were noted at 7 days post RA treatment that coincided with elevated levels of bioavailable RA (Figure 2D). Afterward, decreases were observed following the withdrawal of RA despite levels not returning to baseline until the last day of the study. Consequentially, these data partially support the findings from the proceeding in vitro studies shown in Figure 4A,B. However, our observations need to be confirmed with a large number of animals undergoing cART.

### 3.3. Retinoic Acid (RA) Up-Regulates α4β7 Expression Levels and Induces a Distinct Immune Activation Profile

Obtaining a perspective of RA addition could improve measures of the viral reservoir, therefore we tested the effects of the addition of RA on T cell phenotype (activation/memory and differentiation status). As described in Figure 3A, we determined the frequencies of α4β7^hi^ expressing CD4^+^ T cells following in vitro culture with various stimulating agents. As evident in Figure 3A,B, there were no differences in the frequencies of α4β7^hi^ expressing CD4^+^ T cells regardless of the stimulating agents when CD3/28 beads were not added. Nevertheless, cell cultures with media control showed values of 22.85% (4.2–34.2%). media containing IL-2 showed frequencies of 21.9% (7–24.6%), and media containing IL-2 + RA showed frequencies of 20.5% (7.41–26.5%) α4β7^hi^ expressing CD4^+^ T cells. However, when, PBMCs were cultured in the presence of anti-CD3/CD28 beads, the frequencies of CD4^+^ T cells expressing α4β7^hi^ increased to 36.5% (1.85–40%). The addition of IL-2 had a minimal effect on α4β7^hi^ expression yielding values of 38.58% (12.9–42%).

However, the addition of RA to the anti-CD3/CD28 + IL-2 + RA-treated cells produced a significantly (*p* < 0.05) higher number of α4β7^hi^ expressing CD4^+^ T cells, 91.98% (54.6–99%) compared to those cultured in the presence of anti-CD3/CD28 beads and IL-2 (Figure 3B). Therefore, the addition of RA to anti-CD3/CD28 activated PBMCs cultured in IL-2 enriched media led to a statistically significant upregulation of α4β7^hi^ expression on CD4^+^ T cells (*p* < 0.0001) (Figure 3B).

The anti-CD3/CD28 + IL-2 + RA cultured cells were further examined for their expression of activation markers (Appendix A) and markers of memory/differentiation (Appendix A). For this analysis, we examined these markers on the gated (CD4^+^/CD3^+^/lymphocytes/single cells) population of Total, α4β7^+^ and α4β7^−^ CD4^+^ T cells. The results also revealed that in anti CD3/CD28+ IL2 + RA cell cultures, α4β7^+^ and α4β7^−^ CD4^+^ T cell populations have distinct immune activation profiles. Thus, the α4β7^+^ CD4^+^ T cells when compared with α4β7^−^ CD4^+^ T cells had increased percent frequencies of CD25 (21.6 (17.8–23.9) vs. 13.6 (12–20.4), (*p* = 0.0097)) and CD38 (86.4 (79.3–90.2) vs. 73.1 (57.6–75.7), (*p* = 0.0024)). Conversely, α4β7^+^ CD4^+^ T cells had lower frequencies of CD69 expressing CD4^+^ T cells (11.3 (9.28–12.6) vs. 17.7 (13.5–19.4), (*p* = 0.0052)) and a reduced frequency of PD-1 expressing CD4^+^ T cells (21.4 (11.6–27.8) vs. 32.4 (23.5–47.9), (*p* = 0.0064)). There was no statistically significant difference in the expression of CTLA4 and Ki67 immune activation markers between the two cell subsets (Figure 3C). Surprisingly in anti-CD3/CD28 + IL-2 cell culture conditions, only CD38 remained significantly elevated in α4β7^+^ versus α4β7^−^ CD4^+^ T cells (10.5 (3.78–32.6) versus 3.02 (1.28–10.3))% (*p* = 0.0317). No significant changes were observed in %CD25 ((7.8 (6.4–9.9) versus 9.5 (7–10.1)), %CD69 (0.47 (0.03–0.76) versus 0.26 (0.042–0.49)), %CTLA4 (4.16 (2.11–6.16) versus 4.03 (0.34–1.1)), %Ki67 (0.58 (0.34–1.1) versus 0.37 (0.21–0.51)) and %PD-1 (3.88 (2.79–5.43) versus 9.03 (6.48–16)) in α4β7+ as opposed to α4β7^−^ CD4^+^ T cell populations (Figure 3D). In Total CD4^+^ T cells, 21.5 (17.4–23.7)% CD25, 83.9 (76.2–88.3)% CD38, 12.9 (10.1–14.1)% CD69, 36.7 (33.5–38.7)% CTLA4, 2.88 (2.03–4.46) % HLA DR, 1.04 (6.47–14.7)% Ki67, 23.9 (12.9–30.9)% PD-1 expression was observed (Figure 3E).

Next, we assessed the expression of markers associated with memory/differentiation. Upon CD3/CD28 + IL2 + RA treatment, Total CD4^+^ T cells had 42.7 (37.5–56.8) central memory cells (CD45RA^−^ CCR7^+^) (CM), 28.1 (18.5–38.9), (CD45RA^+^ CCR7^+^) naive cells, 16.8 (11–21.6) effector memory (CD45RA^−^ CCR7^−^) (EM) and 9.78 (2.18–19.1) terminal effector memory cells re-expressing CD45RA (TEMRA) subsets (Appendix A). The predominant cell subsets observed within α4β7^+^ CD4^+^ T cells were those of the naive and CM (Figure 4A) phenotype. Alternatively, α4β7^−^ CD4^+^ T cells are mainly comprised of CM and EM CD4^+^ T cells. Memory subset comparisons between α4β7^+^ and α4β7^−^ CD4^+^ T cells revealed that α4β7^+^ CD4^+^ T cells had higher frequencies of naive cells in comparison to α4β7^−^ CD4^+^ T cells (*p* = 0.0005). On the other hand, the α4β7^+^ CD4^+^ T cells had reduced frequencies of EM, (*p* = 0.0008) and TEMRA, (*p* = 0.0008) in comparison to α4β7^−^ CD4^+^ T cells (Figure 4A). Remarkably, although the frequencies of CD4^+^ EM and CD4^+^ TEMRA were lower in α4β7^+^ CD4^+^ T cells (Figure 4A), there was an increased expression of CD10 T cell-specific apoptosis marker on α4β7^+^ CD4^+^ TEMRA cells (*p* = 0.0076) (Figure 4C). Thereafter, we sought to understand whether changes in memory and differentiation profiles of CD4^+^ T cells were driven by RA. To address this, we profiled the CD3/CD28 + IL-2 culture conditions. This revealed that the predominant phenotype in α4β7^+^ CD4^+^ T cells was TEMRA whilst α4β7^−^ CD4^+^ T cells mainly contained EM cells (Figure 4B). α4β7^+^ vs. α4β7^−^ group-wise comparisons showed that α4β7^+^ CD4^+^ T cells also maintained elevated levels of naive T cells compared to α4β7^−^ CD4^+^ T cells (*p* = 0.0085). As observed in RA cell culture conditions, there were also similar reductions in the frequencies of effector memory CD4^+^ T cells in α4β7^+^ CD4^+^ T cells in comparison to α4β7^−^ CD4^+^ T cells (Figure 4B). However, in contrast to anti-CD3/CD28 + IL-2 + RA culture conditions, CD10^+^ levels were increased in α4β7^−^ as opposed to α4β7^+^ CD4^+^ T cells (Figure 4D)). Lastly, we sought to confirm whether the differences in immune activation of CD3/CD28 + RA-treated cells reported in Figure 3D were due to changes in memory cell subsets noted in Figure 4A (Appendix A).

Remarkably, there was an increase in CD38^+^ expression within naive α4β7^−^ as opposed to α4β7^+^ CD4^+^ T cells (*p* = 0.0161). In contrast, CM (*p* = 0.0002), EM (*p* = 0.0002) and TEMRA (*p* = 0.0054) α4β7^+^ CD4^+^ T cells had elevated levels of CD38^+^ expression in comparison to α4β7^−^ CD4^+^ T cells (Figure 4E). Lastly, we detected elevated levels of PD-1 expression in naive α4β7^−^ compared to α4β7^+^ CD4^+^ T cells (*p* = 0.0031) (Figure 4F).

## 4. Discussion

The major impediment facing the success of a variety of diverse HIV cure interventions is a lack of an accurate method that can reliably measure the size of the viral reservoir. Routine methods like QVOA require more than one round of activation in order to maximize in vitro activation of all cells harboring latent provirus [44,45]. To improve the sensitivity of these assays, a variety of T cell stimuli has been tested for potential inducers of improved T-cell activation. Amongst the stimuli tested, CD3/CD28 antibodies were found to possess the greatest T-cell activation potential despite not fully reactivating the viral reservoir [46]. Thus, we sought to test the effect of supplementing RA to immune activation stimulants such as anti-CD3/CD28 beads and PMA/ionomycin that are routinely used in standardized QVOA/TILDA protocols [30]. This approach was taken with the anticipation that we would maximize levels of cell activation and improve the sensitivity of assays that are used to quantify the size of the viral reservoir.

By using TILDA, we confirmed the ability of RA to enhance the re-activation of the replication-competent viral reservoir in peripheral rhesus macaque CD4^+^ T cells following activation with PMA/ionomycin. The findings were consistent with what we observed within lymph node derived lymphoid CD4^+^ T cells as RA addition to PMA/ionomycin led to an increase in the detection of ms RNA per million CD4^+^ T cells. Further, in QVOA, increased reactivation of the replication-competent latent virus was observed upon the addition of RA, further revealing the promising potential of this molecule to increase IUPM estimates while performing this assay. Our observation that RA is capable of reactivating replication-competent latent SIV is similar to the findings of Li et al., who showed that acitretin, an FDA-approved homologue of RA also leads to increased transcription of activated HIV-1 provirus [47]. Likewise, Zhang et al. showed that supplementation of RA during QVOA improves the reactivation of the latent reservoir in sorted human memory CD4^+^ T cells [21].

Improved recovery of the latent virus during QVOA and TILDA has been associated with increased activation of the viral reservoir. Recent findings by Nawaz et al. indicated that RA was capable of inducing CD4^+^ T cell activation by acting as a second co-stimulatory signal during T cell receptor (TCR)-mediated activation [48]. Thus, we next sought to understand how the treatment of PBMCs with RA following activation with CD3/CD28 beads affects the extent of immune activation/differentiation of CD4^+^ T cells. The addition of RA resulted in the increased expression of α4β7 on CD3/CD28 activated CD4^+^ T cells in vitro. Several others have also shown that the in vitro stimulation of CD4^+^ T cells with RA enhances surface expression of α4β7 [48,49].

Secondly, our results reveal that in comparison to α4β7^−^ T cells, α4β7^+^ T cells expressed elevated levels of CD38 and CD25 coupled with reduced CD69 and PD-1 levels. No changes in levels of the proliferation/cell cycling marker Ki67 were noted. As such, α4β7^+^ T cells expressed lower levels of the early immune activation marker (CD69) and showed increased responsiveness to IL-2 through CD25 (IL-2Rα) together with elevated expression of the cyclic ADP hydrolysis catalyzing ectoenzyme (CD38) during the later stages of T-cell receptor-mediated activation [50,51,52].

The reduced expression of PD-1 noted on α4β7^+^ T cells could be indicative of lower levels of end-stage activation/during the terminal stages of immune activation and not immune exhaustion since there is no accompanying change in the other studied immune checkpoint blockade inhibitor, CTLA4 [53]. Recent reports indicate that transient expression of PD-1 without the co-expression of multiple immune checkpoint blockade markers like CTLA4, TIGIT, Tim-3, Lag-3, 2B4 and CD160 is reflective of immune activation and not an exhausted state [54]. These findings support the argument that RA supplementation activates SIV latent cells that have been previously described to possess multiple immune checkpoint blockade markers [55] by providing extra α4β7 and RA co-stimulation that favors enhanced mid-stage functional immune activation. The reduction of PD-1 levels on the surface of the expanded α4β7^+^ subset further shows how RA could activate cells harboring the latent reservoir since anti-PD-1 molecules have been reported to reverse latency [56].

Next, RA-induced α4β7^+^ CD4^+^ T cells also possessed distinct memory and differentiation profiles that were mainly characterized by an expansion of naive CD4^+^ T cells and lower levels of both EM and TEMRA CD4^+^ T cells in comparison to α4β7^−^ CD4^+^ T cells in vitro. In agreement with our findings, Ding et al., also reported the use of immobilized CD3 beads followed by treatment with RA that generated a similar distribution of α4β7^+^ CD4^+^ T cell subsets that was characterized by a predominantly naive CD4^+^ T cell phenotype [57]. Earlier reports by Cutrona et al. indicated that CD10 expression on T cells could be used to reliably identify cells undergoing apoptosis during cell division [58,59]. Unsurprisingly, we observed elevated expression of CD10 within the TEMRA CD4^+^ T cells and an increasing trend within the EM CD4^+^ T cells of RA-induced α4β7^+^ CD4^+^ T cells hence accounting for their rapid turnover.

Furthermore, in conditions without RA treatment, the reduced frequencies of naive CD4^+^ T cells alongside a predominant CD4^+^ TEMRA population within the α4β7^+^ vs. α4β7^−^ population revealed that naive T cells were the CD4^+^ T cell subset that was most responsive towards the addition of RA. This was further corroborated in our in vivo rhesus macaque model where a strong positive correlation between the bioavailability of RA and the frequency of naive α4β7^+^ CD4^+^ T cells was detected. Increased expression of CD38 across CM, EM and TEMRA cells shows that supplementation with RA increases immune activation across multiple cell subsets. Lastly, the reduced expression of PD-1 within the naive CD4^+^ T cell subset probably suggests that SIV reservoirs could be most affected within this cell phenotype.

Our findings show that supplementation with RA to promote additional immune activation through the α4β7 pathway provides another option that could be explored to increase reactivation of the viral reservoir. It is possible that RA reactivation of the viral reservoir and subsequent viral replication involves LFA-1 activation within the α4β7 pathway as this molecule has been shown to be crucial for viral replication [60]. However, this needs to be validated in future studies aimed at understanding the contribution of diverse transcription factors towards SIV persistence.

The direct link between α4β7 expression and the viral reservoir is supported by recent findings by Uzzan et al. It was recently showed that administering the Food and Drug Administration (FDA)-approved drug Vedolizumab that masks α4β7 expression in leukocytes reduces the size of lymphoid aggregates in the terminal ileum of HIV infected individuals [61]. Lymphoid aggregates have been suggested as key foci in which the latent reservoir is maintained within the gut. By reducing α4β7 surface expression on leukocytes, they observed that reducing α4β7 surface expression led to a reduction in naive CD4^+^ T cells and a decrease in CD4^+^ T cell activation. These findings support our observations where we similarly noticed that increasing α4β7 expression through RA supplementation results in the expansion of naive CD4^+^ T cells and increased global CD4^+^ T cell activation.

Furthermore, the RA derivative Acitretin was reported to disrupt the HIV reservoir by promoting apoptosis of latently HIV-infected cells through activation of the RIG-I signaling pathway [47]. We observed parallels in CD4^+^ T cells from RA-stimulated cultures where the apoptosis marker CD10 was elevated in α4β7 expressing effector and TEMRA cells. Future studies are required to understand and validate the interdependence between α4β7 expression, RIG-1 signaling and stability of the viral reservoir as some study groups argue that Acitretin does not cause optimal activation and subsequent apoptosis of latently-infected cells [62].

Previous studies have suggested that the presence of retinoic acid-responsive elements (RARE) within viral promotor regions (long terminal repeats) may highlight a direct role of RA in ex-vivo upregulation of ms-RNA expression from lymphoid latent reservoirs of HIV/SIV, as described in our modified TILDA experiment [63,64]. This needs to be further confirmed by functional assays using modified viral promoters having defective or deleted RARE sequences including the role of the transcription factors, retinoic acid and retinoid X-receptors [42]. Additional studies are also warranted to confirm whether the observed enhanced reactivation of the latent reservoir arises from the ability of RA to activate provirus integrated into sites that are not responsive to CD3/CD28 activation as opposed to enhanced viral multiplication of reactivated provirus.

Further, we observed enhanced naive cell subsets with the α4β7^+^ fraction of CD4^+^ T cells that were accompanied by increased reactivation of the viral reservoir, emphasizing that the need for latency [42] central memory cells as the viral reservoir could be heterogeneously distributed within the diverse CD4^+^ T cell subsets [65]. This finding is supported by Zerbato et al., who recently reported that indeed naive CD4^+^ T cells harbor replication-competent virus, albeit at levels lower than central memory CD4^+^ T cells [66]. In addition, it has been reported that long-lived naive CD4^+^ T cells contribute to the persistent HIV-1 reservoir and are maintained through homeostatic proliferation. Worse still, HIV-1 proviruses contained within this cell subset are less prone to reactivation using only conventional T-cell receptor stimulants [67]. The use of additional stimulants like RA together with conventional TCR stimulants could improve measures of the viral reservoir more so in cell subsets like naive CD4^+^ T cells that contain proviruses that are less compelled to reactivation.

It should also be noted that the viral reservoir is highly heterogenous and exists in a continuum [68,69]. Unfortunately, routinely used assays such as QVOA and HIV DNA assays only provide quantification for less than 1% of the proviruses that are susceptible to reactivation [45]. In addition following effective ART, the treatment reservoir has been characterized to comprise diverse cells such as myeloid cells [70] and a small fraction of resting naive CD4^+^ T cells [67,71] whose quantification remains underexplored. Future viral reservoir studies should continue to be redirected towards improving the activation potential of commonly used stimulants and enhancing the sensitivity of routinely used assays. There is also a need for additional studies aimed at understanding the composition of the replication-competent viral reservoir within sorted α4β7^+^ CD4^+^ T cells and accompanying subsets in different tissue compartments. This is important because there are differences in α4β7^+^ expression in CD4^+^ T cells with higher frequencies observed within the gut in comparison to blood and various lymph nodes, except mesenteric lymph nodes [72,73]. Understanding how these differences will affect the measures of the viral reservoir following RA supplementation remains to be fully evaluated.

Although we investigated the utility of RA to enhance virus replication in RMs that were not virologically suppressed, we provide a glimpse into how RA could be used to supplement readouts in in vivo assays to study latency. Dosing with RA could be crucial in adoptive transfer experiments where diverse sorted latently-infected cells could be infused into recipient animals and readouts of active virus replication, subsequently being used to confirm the presence of replication-competent reservoirs. Considering all this, our results indicate that the addition of RA whilst measuring the size of the viral reservoir provides an avenue that can be used to provide higher endpoint readouts by improving cell activation and enhancing the dynamic range of commonly used SIV/HIV latency assays.

## Figures and Tables

**Figure 1 cells-09-02076-f001:**
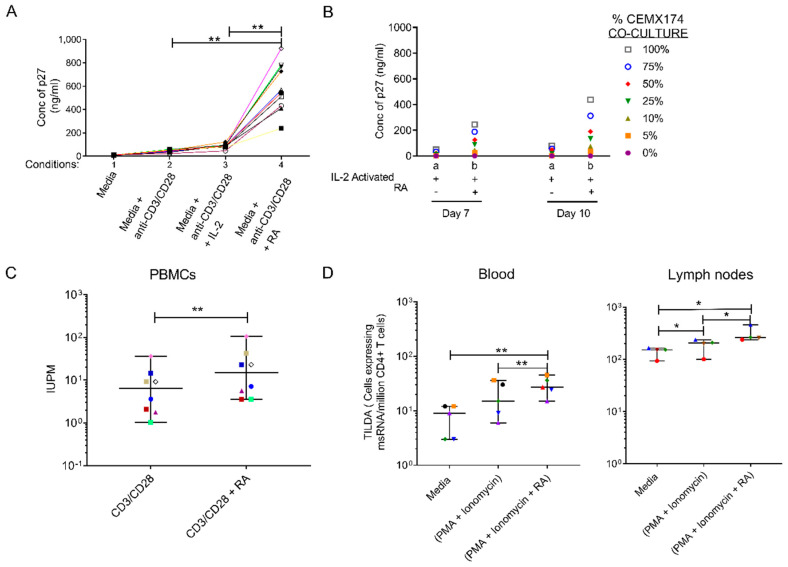
Retinoic Acid (RA) enhances viral replication in vitro and improves detection of the viral reservoir. (**A**) P27 levels were measured in supernatant of 10 randomly selected naive Peripheral blood mononuclear cells (PBMCs) infected with SIVmac239 after in vitro culturing with (1) media, unstimulated (2) anti-CD3/CD28 beads to activate the T-cell receptor (3) anti-CD3/CD28 + IL-2 and 4) anti-CD3/CD28 beads + IL-2 together with RA added under separate conditions on day 10. (**B**) Levels of p27 detected in supernatant fluids from pooled (*n* = 10 RMs) enriched CD4 T cells that were treated with either (a) anti-CD3/CD28 and IL-2 only or (b) anti-CD3/CD28 + IL-2 and RA. Thereafter, viral expansion was carried out by the co-culture with CEMX174 cells across different concentrations (0 to 100%) on days 7 and 10 respectively. (**C**) Quantitative Viral Outgrowth Assay (QVOA) of CD4^+^ T cells purified from PBMCs of anti-retroviral therapy (ART)-suppressed macaques indicating levels of Infectious Units per Million (IUPM) in CD3/CD28 versus CD3/CD28 + RA conditions (*n* = 8). (**D**) levels of msRNA transcripts obtained from enriched CD4 T cells collected from PBMC (*n* = 5 RMs) or auxiliary lymph node CD4^+^ T cells (*n* = 4 RMs) that were cultured in media only, media plus phorbol myristate acetate (PMA) + ionomycin and media plus PMA + ionomycin supplemented with retinoic acid. * shows *p* < 0.05 and ** represents *p* < 0.001 significant difference across studied groups obtained using Wilcoxon matched pairs signed rank tests.

**Figure 2 cells-09-02076-f002:**
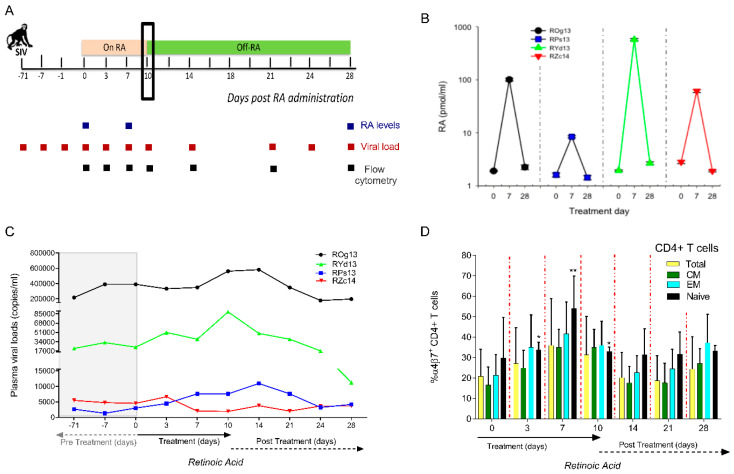
Administration of RA affects SIV replication kinetics by causing modest increases in peripheral SIV replication and the expansion of naive CD4^+^ T cells (*n* = 4 RMs). (**A**) Pictorial schema indicating that RMs were provided with a daily regimen of 10 mg/kg of all trans RA orally for 10 days and then followed for 28 days. (**B**) Levels of RA in plasma were measured at baseline, on day 7 of RA treatment and day 28 (18 days post RA treatment). (**C**) Plasma viral loads measurements were carried out at prospective timepoints prior to and following the administration and later withdrawal of RA. (**D**) Prospective box and whisker plots showing differences in total CD4^+^ T cells and CD4^+^ T cell subsets. CD4^+^ T cell phenotypes were grouped based on their CD28 and CD95 expression. Naive CD4^+^ T cells were categorized as CD28^+^/CD95^−^, central memory CD4^+^ T cells as CD28^+^/CD95^+^ and effector memory CD4^+^ T cells as CD28^−^/CD95^+^.

**Figure 3 cells-09-02076-f003:**
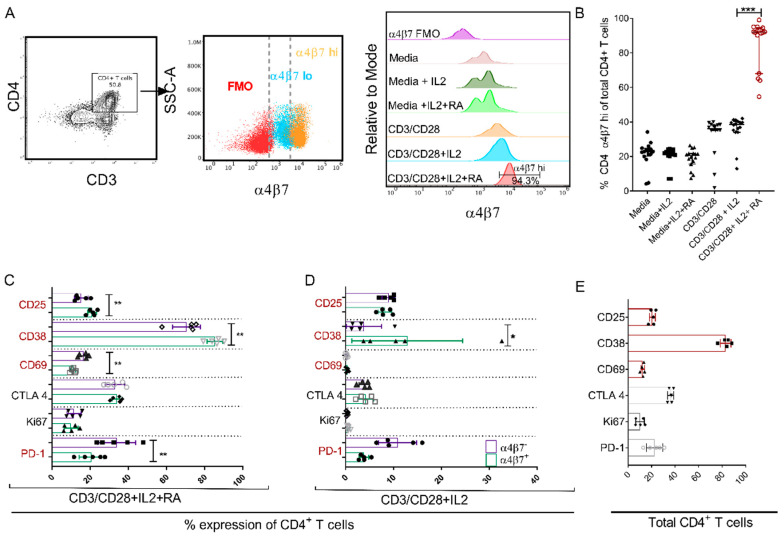
The addition of RA upregulates α4β7 hi expression on CD4^+^ T cells cultured under different conditions resulting in distinct immune activation profiles between α4β7^+^ and α4β7^−^ CD^4+^ T cells. Flow cytometry gating strategy showing: (**A**) CD4^+^ T cells derived from CD3^+^/lymphocyte/live and single cells and later profiled for the extent of α4β7 expression. Following this is an overlay dot plot showing the spread of α4β7 (Fluorescence minus one (FMO)—red, α4β7 lo—blue and α4β7 hi—orange) within the CD4^+^ population. Then, adjacent to this is a representative overlay histogram generated after flow cytometry analysis indicating levels of α4β7^hi^ expression on CD4 T cells normalized to mode following the culturing of PBMCs under the different conditions: media only, media + IL-2, media + IL-2 + RA, anti-CD3/CD28, anti-CD3/CD28 + IL-2, anti-CD3/CD28 + IL-2 + RA and α4β7 FMO as a control. (**B**) Aligned dot plot showing the frequency of CD4 + α4β7^hi^ T cells in total CD4^+^ T cells obtained from 16 naive RMs and cultured under different conditions named in (**A**). (**C**) Percent expression of differences in IA markers CD38, CD69, CTLA 4, Ki67 and PD-1 expressed on α4β7^+^ vs. α4β7^−^ CD4^+^ T cells after cell culture in media comprising of anti CD3/CD28 beads, IL-2 and RA (*n* = 5 rhesus macaque PBMCs) and (**D**) α4β7^+^ vs. α4β7^−^ CD4^+^ T cells following cell culture media containing anti CD3/CD28 beads and IL-2 (*n* = 5 rhesus macaque PBMCs). (**E**) Levels of percent expression of several immune activation (IA) markers (Ki67, CD25, CD38, CD69 and PD-1) on total CD4^+^ T cells. * shows *p* < 0.05, ** indicates *p* < 0.001 while *** denotes *p* < 0.0001 resulting from Wilcoxon matched pairs signed rank tests between paired groups of the different comparisons.

**Figure 4 cells-09-02076-f004:**
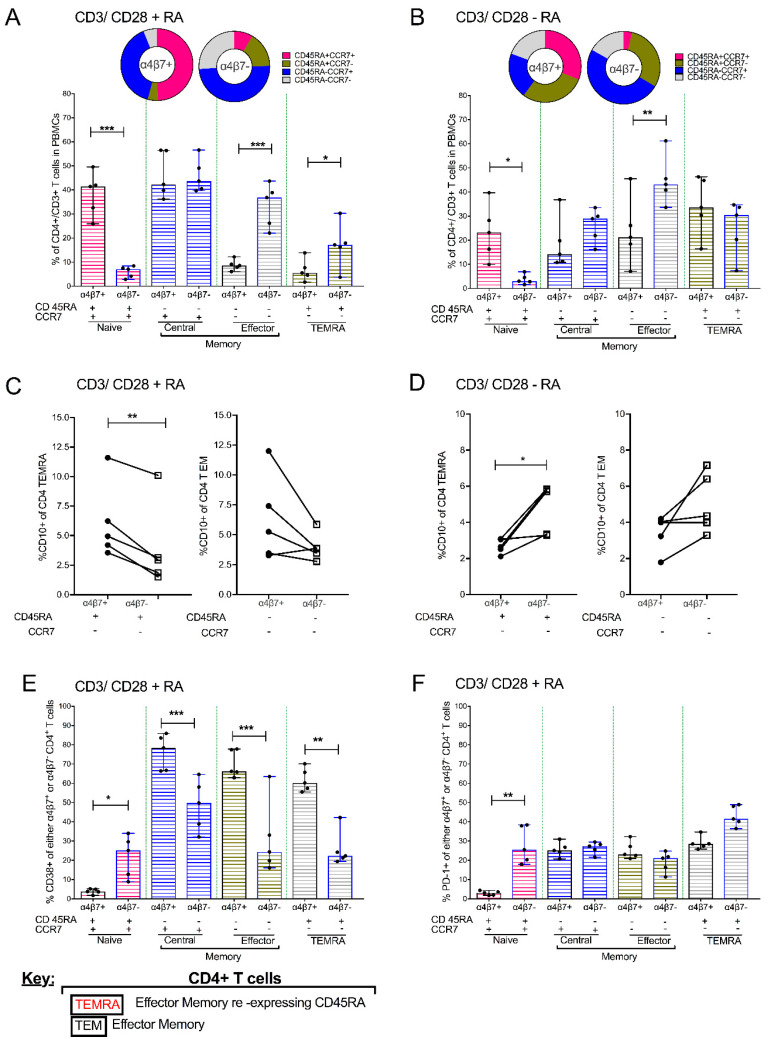
Addition of 1uM RA to PBMCs that were previously activated with anti-CD3/CD28 beads and cultured in IL-2 enriched media leads to different memory/differentiation profiles in α4β7^+^ vs. α4β7^−^ CD4^+^ T cells (*n* = 5). Combined bar graph and pie chart showing the comparative distribution of naive, central memory, effector memory (EM) and terminal effector memory cells re-expressing CD45RA (TEMRA) subsets within α4β7^+^ vs. α4β7^−^ CD4+ T cells obtained from PBMC treated with (**A**) CD3/CD28 beads + RA or (**B**) CD3/CD28 beads-RA. (**C**) Dot plots indicating paired comparisons for the quantification of CD10+ levels in α4β7^+^ vs. α4β7^−^ partitions of CD4^+^ TEMRA cells and CD4^+^ TEM cells in CD3/CD28 + RA or (**D**) CD3/CD28 − RA treatment conditions. (**E**) The distribution of CD38 and (**F**) PD-1 markers of immune activation amongst CD4^+^ T cell subsets obtained from PBMC treated with CD3/CD28 beads + RA conditions. * shows *p* < 0.05, ** indicates *p* < 0.001 while *** denotes *p* < 0.0001 obtained using Wilcoxon matched pairs signed rank tests.

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
