# Peer review of "Retinoic Acid Improves the Recovery of Replication-Competent Virus from Latent SIV Infected Cells"

_cells, 2020, doi:10.3390/cells9092076_

Round 1

Reviewer 1 Report

In order evaluate whether the in vitro/in vivo addition of retinoic acid enhances virus replication and improves the detection of latent virus, Olwenyi and colleagues culture PBMC from uninfected and SIV infected rhesus macaques with anti-CD3/CD28 + IL-2 in the presence or absence of retinoic acid. They subsequently analysed viral load and p27 expression by RT-PCR and ELISA and the viral reservoir by QVOA and TILDA. They furthermore assessed α4β7-expression levels, memory/differentiation status and immune activation by flow cytometry.

The addition of RA led to an increase in α4β7high expression on activated CD4+ T cells and enhanced production of viral particles in ex-vivo cultures of PBMCs obtained from ART-treated SIV-infected macaques. Plasma from monkeys orally treated with RA for 10 days caused a “modest increase in peripheral SIV replication”.

The authors conclude that RA could be used to enhance current in vitro stimulation protocols to determine latent reservoirs more efficiently.

The authors conclusively demonstrate that the production of viral particles as indicated by p27 expression is enhanced in samples stimulated with CD3/CD28 and retinoic acid compared to unstimulated or samples solely stimulated with CD3/CD28. This could indeed help to improve assays that assess the size of the latent reservoir.

However, the manuscript needs to be streamlined and rearranged: how are the distinctive differentiation and activation profiles of α4β7+ vs α4β7- cells (Figure 2) relevant to the main statement of the manuscript? Two topics seem to be mixed in a non-conclusive way: a) a4b7 and b)latent reservoir. How big is the proportion of a4b7+cells to the reservoir? Will their specific activation make a marked difference in the analysis of PBMC or lymph nodes?

Author Response

Attached the response letter  

Reviewer 2 Report

In this manuscript, Olwenyi et al. aimed to study whether the addition of retinoic acid (RA) enhances virus replication and improves detection of latent virus. They found that RA increase viral replication in in vitro and in some animals in vivo. QVOA and TILDA assays indicated that RA may augment reactivation of the replication competent viral reservoir in peripheral blood from ART-suppressed macaques  The authors augured that RA can be a useful approach to enhance the efficiency of current protocols used for in vitro estimates of CD4+ T cell latent reservoirs. While it is interesting the RA can boost CD3/CD28 induced activation or reactivation of HIV transcription, the concept of use RA as a potential LRA has been shown before. This may reduce the novelty of this study. In addition, the experimental design of animal study is little confused as it is disconnected from this study as a whole.

Major points:

  1. In the 1st sentence, in addition to Berlin patient, there is at least one more patient which can be considered as “cure” by therapy of bone marrow transplantation.
  2. The molecular basis of RA signaling is probably involved in many other pathways (not just α4β7), such as RIG-I signaling, please discuss and explain why it was not included in this study. There is an argument upon whether activation of RIG-I signaling by RA derivative is able to disrupt latent HIV.
  3. In Figure 1., can the authors include transitional memory cells (TM) in their T cell phenotype study since TM cells are known HIV reservoirs?
  4. Many data have been collected in vitro, I am wondering why the authors did not examine the T cell phenotype changes and immune activation/suppression markers in the in vivo study (Figure 4), which is much more relevant than in vitro study (Figures 1-2).
  5. In Figure 3, the co-culture of CD4+ T cells with CEM-174 did not increase the p27 levels when stimulated with CD3/CD28 beads. I am not sure whether CEM-174 was included in the Figure 3C-D.
  6. The animal study seems not necessary in this study since these animals were not under ART. The virus kinetic data had no clue of reservoirs seeding or reactivation in vivo. Actually, it is a good opportunity to determine the immune cell phenotype when RA was administrated in vivo.

Author Response

Attached reviewers response 

Reviewer 3 Report

Brief summary: The study presented in the manuscript by Olwenyi et al. aimed to investigate usability of retinoic acid (RA) as an agent to promote reactivation of the latent provirus for the purpose of improvement of existing reservoir quantification assays. The first set of experiments focused on detailed phenotypic characterization of CD4+ T cells following treatment with RA in vitro, focusing on integrin a4b7 expression, activation patterns and maturation subsets. The second set of experiments quantified SIV replication (ELISA, co-culture assays) and induced reservoirs (TILDA, qVOA) with and without treatment with RA ex vivo. Finally, the last datasets presents a pilot in vivo evaluation of the effect of oral RA intake on viral loads in macaques.

Broad comment: Utilization of RA as an agent to improve SIV reactivation, especially in a setting where all phenotypic subsets of T cells (rather than only memory cells) are considered, is the novel aspect and a strength of the presented study. The rigor of research presented is reflected by using several approaches (in vitro, ex vivo, in vivo), as well as providing sufficient methodological details. The strength of the study is not only demonstration that RA works, but providing some mechanistic evidence about how it may work. However, the manuscript is structured in a manner that the main message is buried among facts and experiments, without providing a clear rationale why these experiments were performed. There is no discussion of the recent paper by Zhang et al. “Improving HIV outgrowth by optimizing cell-culture conditions and supplementing with all-trans retinoic acid” Front Immunol. 2020. The authors could present their findings in a much more exciting manner to make the study stand out and its significance and strengths emphasized.

Specific comments:

Introduction:

1). The entire Introduction is presented as a single paragraph. The purpose of the study is not presented. The manuscript would benefit from presenting, in separate paragraphs, a). the need for improving reservoir quantification assays, b). the existing assays and recent attempts to improve them, c). introduction of RA as an agent that can be used to improve the assays, d). the aims of the present study.

2). The logic of why RA should be tested for improving reactivation is eluding in the Introduction. A lot of emphasis is put on studies that show that RA increases permissiveness of cells to infection, not reactivation of the latent reservoir.

3). The first several sentences regarding sterilizing cure take the reader away from the main focus of the manuscript. They can be boiled down to 1-2 sentences stating that there are only 2 cases of sterilizing cure (reference); thus, research still focuses on functional cure (reference), and therefore, accurate quantification of reservoir remains an important problem.

Methods:

1). Section 2.2: This section does not talk about flow cytometry, but “flow cytometry” is present in the title. There is a separate flow cytometry section below.

2). Section 2.2: How were cell viabilities quantified?

3). Please elaborate on the statistical analysis. Was Wilcoxon matched pairs signed rank test used for only flow cytometry, or all data types? If only flow cytometry, how were other datasets analyzed (e.g. ELISA)?

4). Suggestion: separate long sections that contain several methods into paragraphs by method, so that it is easier to follow. For example, section 2.5 is too long and can be broken into paragraphs containing methods for SIV replication, viral expansion and reservoir quantification assays.

Results:

1). The two main conclusions of the experiments that were presented are that: a). RA may contribute to proviral reactivation by promoting activation of a subset of immune cells through integrin a4b7 pathway; b). RA may be working directly on provirus, as evidenced from results of TILDA experiments. Setting up rationale for the experiments in each of the experimental sections would help guide the reader both with respect to why experiments were performed and how the authors arrived at their conclusions. Some rationale is presented in Discussion, but it would be helpful if it were presented up front.

2). Are there any data on the immune cell activation phenotypes in vivo? Adding these data would really strengthen the “immune activation” conclusion of the study.

3). All result sections are presented as long paragraphs. It would be better to break them into paragraphs when a description of the new experiment starts. For example, section 3.2 could be broken into two paragraphs: a). about SIV replication in vitro; b). about reactivation of the latent reservoir.

4). Figure 1D: Heat map may not be the best presentation of these data. It would be better to show bar plots with individual data points as in Figure 2A,B. This would help reduce some numbers from the text, which are hard to follow, and be consistent in presentation with Figure 2.

Discussion:

1). Discussion has a lot of text that should be or already is in Introduction and Results. For example, the first paragraph is very introductory, while the second reiterates results, with some references to previous reports with similar observations. The Discussion could be made shorter, more focused and easier to follow, if all the redundant text is cleaned out and the section is focused on what the results mean and why they are significant for the field.

2). Limitations are well presented, but their presentation should be consolidated.

3). A paragraph summarizing the strengths of the study would substantially improve this manuscript. How is this study different from Zhang et al.?

Author Response

Attached Reviewers response

Round 2

Reviewer 2 Report

The authors generally responded to my questions. While this MS could be improved, it can be accepted for publication.

Author Response

We thank this reviewer for accepting our manuscript for publication. As suggested,  we made efforts to improve the manuscript overall by fixing some minor errors. 

Reviewer 3 Report

The authors extensively responded to the reviewers' comments and significantly  improved the manuscript. Unfortunately, it seems that there are some logic omissions that resulted from the re-ordering the paragraphs that need to be addressed. For example, current Results section 3.2 used to be the last section in the original manuscript, and it still starts with the word "finally". Similarly, section 3.3, which was the first section, starts with the introduction "we first determined". The authors should carefully revise their logic and introduce the final edits so that the revised manuscript flows properly. 

Author Response

We thank this reviewer for accepting our revised manuscript. Again our apologies for the oversight. As suggested we have now corrected these issues and made changes to the flow of the manuscript. We thank this reviewer for cathing these important errors and now we have gone over the entire manuscript and streamlined these few errors. Further, we have carefully proofread the manuscript and fixed all the English errors.